# FunKAN: Functional Kolmogorov-Arnold Network for Medical Image Segmentation

## Abstract

Medical image segmentation is pivotal for clinical diagnosis, however, remains challenging due to complex anatomies and imaging artifacts. While deep learning offers powerful solutions, prevailing architectures lack inherent interpretability and often rely on empirically designed components. Kolmogorov-Arnold networks provide a mathematically interpretable alternative but fail to preserve the spatial structure of visual data, as they process flattened feature vectors. To bridge this gap, we introduce Functional Kolmogorov-Arnold Network (FunKAN), a novel framework that generalizes the Kolmogorov-Arnold theorem to functional spaces. FunKAN parametrizes its inner functions via truncated spectral expansion over Hermite basis functions, enabling direct processing of 2D feature maps within a theoretically grounded, interpretable design. Leveraging this, we integrate FunKAN into the U-shaped architecture, yielding a new state-of-the-art segmentation model across diverse medical imaging modalities. Extensive benchmarks on BUSI (ultrasound), GlaS (histology), and CVC-ClinicDB (colonoscopy) datasets show that U-FunKAN outperforms strong baselines (U-Net, KAN, Mamba), achieving IoU and F1-score improvement and superior efficiency in terms of Gflops. Our work unites theoretical function approximation and practical medical image analysis, offering the novel state-of-the-art solution for clinical applications.

## 1 Introduction

Computer-aided diagnosis systems now constitute essential components of the contemporary medical imaging infrastructure, addressing critical issues such as rising diagnostic workloads and interpreter variability (Kadhim et al., 2022). Nevertheless, their clinical utility is often limited by segmentation accuracy. Although deep learning has revolutionized medical image analysis, prevailing neural architectures frequently rely on empirically derived components that lack theoretical justification and exhibit limited generalization across imaging modalities (Borys et al., 2023). To address these gaps, this research aims to develop a novel, theoretically-grounded deep learning architecture for cancer segmentation across diverse medical imaging contexts.

Extensive clinical evidence confirms that early-stage disease detection and subsequent diagnostic confirmation, whether through ultrasound imaging, histopathological analysis or colonoscopy, correlate strongly with enhanced long-term survival probabilities (Abhisheka et al., 2023). Diagnostic accuracy, however, is compromised by an exponential increase in imaging examinations coupled with a critical shortage of trained specialists, including radiologists and pathologists. Indeed, the World Health Organization (WHO) reported that in 2024 breast cancer affected 2.3 million women worldwide annually, resulting in 670000 deaths. This disease can develop at any age after puberty. Breast cancer outcomes exhibit a significant association with socioeconomic development levels: in very high-HDI nations women face a lifetime breast cancer incidence of 1 in 12 and a mortality rate of 1 in 71. In stark contrast, low-HDI countries demonstrate both a lower incidence rate of 1 in 27 and disproportionately higher mortality 1 in 48, highlighting the substantial deficiencies in early detection capabilities and therapeutic accessibility.

In light of the advancements of Kolmogorov-Arnold networks (KANs) (Liu et al., 2024b) for both medical image enhancement (Penkin & Krylov, 2025) and segmentation (Li et al., 2025), we propose a Functional Kolmogorov-Arnold Network (FunKAN) – a novel extension of the original KANs

to better address fundamental image processing requirements. While the original theorem (Kolmogorov, 1957) applies to continuous functionals $f(x_1, ..., x_n)$ on $\mathbb{R}^n$, we hypothesize its generalization to continuous functionals $f(\chi_1, ..., \chi_n)$ on $H^n$, where each $\chi_i$ states for an element from a Hilbert space $H$. The proposed functional extension enables representation-based feature extraction in a natural way for high-dimensional image latents by obviating feature flattening and considering each 2D feature map $\chi_i$ as an element of an underlying Hilbert space $H$, viewed on a spatial grid $h \times w$. Thus, the proposed approach preserves an intrinsic structure of imaging data and establishes a principled connection between the classical approximation theory and the contemporary deep learning approaches for image analysis. To ensure reproducible research, we release entire codebase on GitHub featuring PyTorch Lightning for modularity, Ruff for code quality enforcement, YAML-based configuration system for experiments management.

Our contributions can be summarized as follows:

1. **Theoretical contribution:** We propose an extension of the Kolmogorov-Arnold theorem onto functional spaces.

2. **Empirical validation:** We introduce Functional Kolmogorov-Arnold Network (FunKAN).

3. **State-of-the-art segmentation quality:** U-FunKAN achieves state-of-the-art segmentation accuracy across three distinct medical imaging modalities:
   - breast ultrasound, BUSI dataset (Al-Dhabyani et al., 2020),
   - histological gland structures, GlaS dataset (Valanarasu et al., 2021),
   - colonoscopy polyp detection, CVC-ClinicDB dataset (Bernal et al., 2015).

## 2 RELATED WORK

### 2.1 MEDICAL IMAGE SEGMENTATION

The advent of deep learning has driven substantial progress in medical image analysis, particularly in the task of semantic segmentation. This technology has transitioned from a research curiosity to an essential tool for the automated and precise delineation of anatomical structures and pathological regions. Driven initially by convolutional neural networks (CNNs), like U-Net, and more recently by vision transformers, these models have demonstrated remarkable capabilities in tumor detection in oncology.

U-Net (Ronneberger et al., 2015) established a foundational encoder-decoder framework with skip connections, facilitating an accurate localization through the integration of high-level semantic information with low-level spatial details. Its widespread adoption in medical image analysis is largely attributable to its robust performance with limited training data. However, the original U-Net architecture exhibits limitations in modeling long-range spatial dependencies and preserving fine structural details.

Early architectural innovations addressed these limitations through several key developments. Attention U-Net (Oktay et al., 2018) enhances feature selectivity through attention gates in skip connections, dynamically emphasizing salient features, useful for a specific task, while suppressing irrelevant ones. U-Net++ (Zhou et al., 2018) improves feature fusion through nested, dense skip pathways, reducing the semantic gap between encoder and decoder features. By aggregating features across multiple scales U-Net++ enhances segmentation quality for anatomically irregular targets (e.g., infiltrating tumor margins), while incurring a greater computational overhead.

Emerging hybrid architectures have introduced novel computational paradigms, like U-Mamba (Ma et al., 2024) and U-KAN (Li et al., 2025). U-Mamba integrates Mamba into the U-Net architecture to capture long-range dependencies with linear computational complexity, making it particularly suitable for high-resolution medical imaging. U-KAN substitutes convolutional layers in the backbone with Kolmogorov-Arnold-motivated adaptive activation functions. The spline-based parameterization of these activations enables more accurate modeling of the complex biological morphologies, particularly irregular tumor margins and vascular networks, while simultaneously addressing the spectral bias (Rahaman et al., 2019) inherent in ReLU-based neural networks. However, the proposed KAN-based backbone processes spatial feature maps as unstructured coordinate collections, thereby ignoring the locality priors essential for image representation. MedKAN (Yang

et al., 2025) and UKAGNet (Drokin, 2024) partially mitigate this limitation through a hybrid convolutional approach, combining adaptive spline-based nonlinearities with spatial inductive biases. However, MedKAN remains constrained by its classification design, lacking any proven generalizations onto medical image enhancement and segmentation pipelines. Whereas UKAGNet does not go beyond the original Kolmogorov-Arnold theorem to adapt the concept further for image-to-image processing pipelines.

## 2.2 Kolmogorov-Arnold Networks

Foundational mathematical theorems provide a powerful starting point for constructing new deep learning architectures, enabling advanced modeling of complex data relationships (Li et al., 2023). A prime example is Kolmogorov-Arnold network (Liu et al., 2024b), which implements the theoretical framework of the Kolmogorov-Arnold theorem through adaptive B-spline embeddings.

The Kolmogorov-Arnold theorem (Kolmogorov, 1957) states that if $f \colon [0,1]^n \to \mathbb{R}$ is a multivariate continuous function, then it can be written as a finite composition of continuous functions of a single variable and the binary operation of addition:

$$f(x_1, ..., x_n) = \sum_{j=1}^{2n+1} \zeta_j \left( \sum_{i=1}^{n} \phi_{ji}(x_i) \right),$$ (1)

where $\phi_{ji} \colon [0,1] \to \mathbb{R}$ and $\zeta_j \colon \mathbb{R} \to \mathbb{R}$ – continuous inner functions of a single variable. The theorem provides another justification that neural networks of sufficient depth and width are capable of forming dense subsets in the space of continuous functions defined over compact domains (Cybenko, 1989).

While the Kolmogorov-Arnold representation theorem offered a theoretically appealing reduction of high-dimensional function approximation to learning univariate functions, the pathological non-smoothness or even fractal character of its inner functions had severely constrained its applications. In 2024 the authors Liu et al. (2024b) presented Kolmogorov-Arnold network, relaxing the original theorem's constraints while preserving its fundamental principles. Unlike the classical representation limited to two nonlinear layers with $(2n+1)$ hidden terms, their architecture permits arbitrary width and depth, leading to the modern differentiable KAN definition:

$$KAN(\mathbf{x}) = (\Phi_L \circ \Phi_{L-1} \circ ... \circ \Phi_1)(\mathbf{x}),$$ (2)

where $\{\Phi_l\}_{l=1}^{L}$ – the Kolmogorov-Arnold layers, defined as:

$$\begin{bmatrix} x_{l+1,1} \\ x_{l+1,2} \\ \vdots \\ x_{l+1,m} \end{bmatrix} = \begin{bmatrix} \phi_{l,11} & \phi_{l,12} & ... & \phi_{l,1n} \\ \phi_{l,21} & \phi_{l,22} & ... & \phi_{l,2n} \\ \vdots & \vdots & \vdots & \vdots \\ \phi_{l,m1} & \phi_{l,m2} & ... & \phi_{l,mn} \end{bmatrix} \begin{bmatrix} x_{l,1} \\ x_{l,2} \\ \vdots \\ x_{l,n} \end{bmatrix},$$ (3)

$$x_{l+1,j} = \sum_{i=1}^{n} \phi_{l,ji}(x_{l,i}),$$ (4)

where $\phi_{l,ji} \colon \mathbb{R} \to \mathbb{R}$ – continuous inner functions, parameterized in a smooth differentiable manner by B-splines.

Recent KAN architectures (Seydi, 2024) have improved efficiency by replacing B-splines with Gaussian radial basis functions (RBFs), resulting in FastKAN (Li, 2024). In ChebyKAN (SS et al., 2024) the authors employ Chebyshev polynomials as a complete orthogonal system to substitute B-splines, achieving enhanced training stability.

Although Kolmogorov-Arnold networks demonstrate powerful capabilities in multivariate function approximation, their naive application to image processing is fundamentally limited. The limitation stems from KANs' treatment of inputs as scalars, thereby ignoring the essential two-dimensional geometric structure inherent in visual data. We resolve this limitation by formulating a functional-space generalization of the Kolmogorov-Arnold theorem, considering each feature map as an element of an underlying Hilbert space $H$. This theoretical advancement motivates our functional Kolmogorov-Arnold network, seamlessly suitable for image processing pipelines by incorporating spatial awareness.

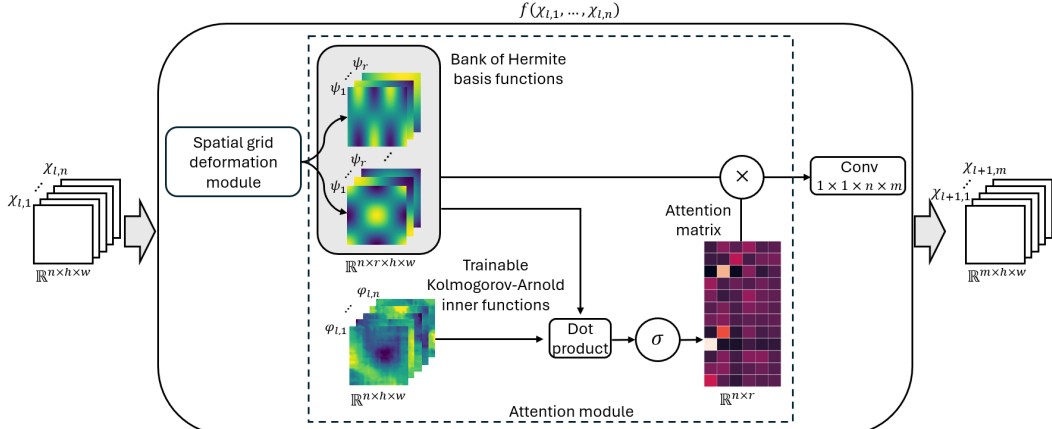

Figure 1: Architecture of Functional Kolmogorov-Arnold Network (FunKAN). The network implements trainable inner functions $\{\varphi_{l,i}\}_{i=1}^n$ through Fourier decompositions over the basis Hermite functions $\{\psi_k\}_{k=1}^r$, where each function is visualized on $h \times w$ spatial grid matching the input feature dimensions, and each decomposition is defined by normalized Fourier coefficients stored in the rows of the attention matrix.

## 3 METHOD

This work introduces a novel neural architecture termed Functional Kolmogorov-Arnold Network (FunKAN), illustrated in Fig. 1. The model is constructed upon a theoretical foundation – a proposed functional generalization of the classical Kolmogorov-Arnold theorem. While the original theorem provides representation for multivariate functions on real-valued inputs, our generalization extends this concept to continuous functionals operating on elements of a Hilbert space:

**Statement 3.1** *If $f$ is a continuous functional on $H^n$, then it can be represented as a composition of linear continuous functionals from the dual space $H^*$, continuous functions of a single variable and the binary operation of addition:*

$$f(\chi_1, ..., \chi_n) \rightsquigarrow \sum_j \zeta_j \left( \sum_i \varphi_{ji}(\chi_i) \right),$$

*where $H$ is a Hilbert space, $\chi_i \in H$, $\varphi_{ji} \in H^*$ and $\zeta_j \colon \mathbb{R} \to \mathbb{R}$.*

The proposed functional extension of the Kolmogorov-Arnold theorem hypothesizes that continuous operators on $H^n$, modeling activation mappings between feature spaces, may be approximated by functionals from the dual space $H^*$.

To construct a computationally tractable model, we ground our approach in the Riesz representation theorem. This theorem establishes an isomorphism between a Hilbert space $H$ and its dual space $H^*$, allowing us to replace the dual element $\varphi_{l,ji}$ with its corresponding representative $\varphi_{l,j}(\chi_{l,i})$ in $H$. We then parameterize these inner functions in a differentiable manner by expressing them via a Fourier expansion truncated to the first $r$ Hermite functions $\{\psi_k\}_{k=1}^r$. Using the fact that Hermite functions are the eigenfunctions of the integral Fourier transform in $L_2(\mathbb{R})$ (Titchmarsh, 1948), the spectral truncation strategy retains the most informative modes and mirrors the frequency-truncation principle in Fourier neural operators (Li et al., 2023). So, the next-layer feature map is computed as:

$$\chi_{l+1,j} = \sum_{i=1}^n \sum_{k=1}^r \langle \varphi_{l,j}(\chi_{l,i}), \psi_k \rangle \psi_k. \tag{5}$$

Thus, we conceptualize each 2D feature map $\chi_{l,i}$ as an element of a Hilbert space $H$, whose values are discretized over the spatial domain $h \times w$. This construction allows for a seamless integration into standard deep learning architectures for image processing, eliminating the necessity of feature flattening.

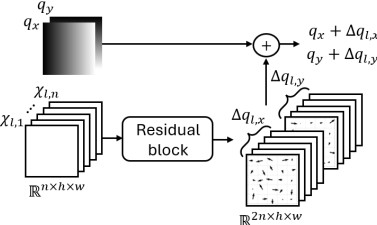

Figure 2: Architecture of the spatial grid deformation module, illustrating learned spatial deformation through residual network-generated offset tensors $\Delta q_{l,x}$, $\Delta q_{l,y}$. These predicted offsets are combined additively with a broadcasted uniform reference grid $\{q_x, q_y\}$ to produce the deformed sampling grid for the basis Hermite functions evaluation.

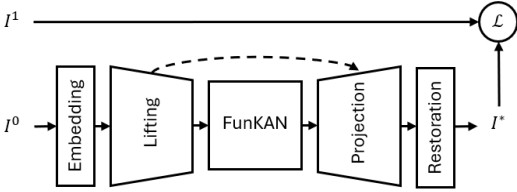

Figure 3: Overview of FunKAN as a multi-purpose backbone for medical image processing. The model processes an input image $I^0$, being supervised by a target image $I^1$ via the loss function $\mathcal{L}(I^*, I^1)$.

The final computational form is derived in two steps. First, the $j$-index is factored out, which is equivalent to a $1 \times 1$ convolution with weights $W_l = \{\omega_{l,j}\}_{j=1}^m$, $\omega_{l,j} \in \mathbb{R}^{1 \times 1 \times n}$. Second, the Hermite basis functions are evaluated on a deformed grid, a strategy inspired by modern implicit architectures (Agro et al., 2024), inducing an additional learnable dependency upon input functions $\{\chi_{l,i}\}_{i=1}^n$. Consequently, the layer output is computed as:

$$\chi_{l+1,j} = \sum_{i=1}^n \omega_{l,j} \left( \sum_{k=1}^r \langle \varphi_l(\chi_{l,i}), \psi_{l,k}(\chi_{l,i}) \rangle \psi_{l,k}(\chi_{l,i}) \right). \tag{6}$$

Hence, a FunKAN layer is parameterized by three core components: a tensor of learnable inner functions $\varphi_l \in \mathbb{R}^{n \times h \times w}$, a set of deformed Hermite basis functions $\psi_l \in \mathbb{R}^{n \times r \times h \times w}$ and convolutional weights $W_l \in \mathbb{R}^{1 \times 1 \times n \times m}$.

The spatial coordinates for the basis Hermite functions evaluation are deformed by a learned vector field $\Delta q_l = \{\Delta q_{l,x}, \Delta q_{l,y}\}$, generated through a residual block (He et al., 2016) (see Fig. 2). Formally, the grid deformation is computed as $q + \Delta q_l$, where $q = \{q_x, q_y\}$ is a uniform grid.

As shown in Fig. 2, the residual block generates spatial offset tensors: $\Delta q_{l,x} \in \mathbb{R}^{n \times h \times w}$ and $\Delta q_{l,y} \in \mathbb{R}^{n \times h \times w}$. These offsets are then added to a uniform grid, yielding the deformed sampling coordinates used for the basis Hermite functions evaluation. Consistent with the pre-activation approach (Duta et al., 2021), the residual block processes activations through batch normalization (Balestriero & Baraniuk, 2022) and ReLU before the subsequent convolution:

$$\Delta q_l = \mathcal{W}_{l,0} * BN(\chi_l) + \mathcal{F}_l(\chi_l), \tag{7}$$

$$\mathcal{F}_l = \mathcal{W}_{l,2} * ReLU\{BN(\mathcal{W}_{l,1} * ReLU\{BN(\chi_l)\})\}, \tag{8}$$

where $\Delta q_l \in \mathbb{R}^{2n \times h \times w}$ – learned spatial offsets, $BN$ – batch normalization and $\{\mathcal{W}_{l,i}\}_{i=0}^2$ – $3 \times 3$ convolutional layers, maintaining the spatial resolution.

While both pre- and post-activation residual architectures possess equivalent theoretical representational capacity, empirical evidence demonstrates superior gradient propagation in pre-activation architectures (Duta et al., 2021). Our implementation ensures stable optimization through batch normalization layers and skip connections.

We leverage FunKAN as a backbone architecture within the U-shaped segmentation framework (see Fig. 3), resulting in the proposed U-FunKAN architecture. The model is configured as follows:

1. Embedding: $3 \times 3$ convolution, projecting an input image into 16-dimensional feature space.

2. Lifting: four consecutive U-Net-like encoder residual blocks with progressively increasing filter count: $32\ (C_1) \rightarrow 64\ (C_2) \rightarrow 128\ (C_3) \rightarrow 128$, – each halving the spatial resolution through strided $3 \times 3$ convolution.

3. Backbone: sequence of three FunKAN blocks, interconnected with skip connections in 128-dimensional feature space ($n = 128$), encapsulating the spectral encoding of each inner function over the first six Hermite basis functions ($r = 6$).

4. Projection: four consecutive U-Net-like decoder residual blocks with gradually decreasing filter count: $128\ (C_3) \rightarrow 64\ (C_2) \rightarrow 32\ (C_1) \rightarrow 16$, – each performing $\times 2$ upsampling via nearest-neighbor interpolation, followed by $3 \times 3$ convolution for feature refinement and skip connection from the corresponding lifting module.

5. Restoration: $1 \times 1$ convolution, projecting ReLU pre-activated features to logits.

The model is trained in a supervised manner using the weighted combination of binary cross-entropy and dice loss:

$$\mathcal{L}_{segm} = \frac{1}{N} \sum_{i=1}^{N} 0.1 \cdot CE(I_i^*, I_i^1) + Dice(I_i^*, I_i^1), \tag{9}$$

where $N$ – batch size, equals to 8.

The Hermite basis is utilized because of the inherent dual localization exhibited by Hermite functions, a property stemming from their role as integral Fourier transform eigenfunctions (Grünbaum, 1982). The number of basis functions ($r = 6$) is determined following the methodology of Penkin & Krylov (2025), which performed a grid search over three candidate bases – B-splines, Chebyshev polynomials and Hermite functions, ultimately selecting six basis functions as optimal.

## 4 EXPERIMENTS

We conducted a comprehensive evaluation of FunKAN on three datasets, selected to validate our method's robustness across anatomical diversity, encompassing oncological, histological and endoscopic structures, and modality variations, including ultrasound, histopathology and colonoscopy.

The experimental framework is implemented in Python 3.12 using PyTorch 2.5, with all models trained and evaluated with full precision on NVIDIA RTX A6000 GPU. The software stack employs PyTorch Lightning 2.5.1, CUDA 11.8 and cuDNN 9. Computational reproducibility is ensured through the seeds setup and YAML-based experiments management. The models were trained from scratch till convergence using Adam (Diederik, 2014) stochastic optimization algorithm ($\beta_1 = 0.9$, $\beta_2 = 0.999$, $\varepsilon = 10^{-8}$) with learning rate manual scheduling upon the scheme: $10^{-4}$, $5 \cdot 10^{-5}$, $10^{-5}$. To enhance generalization, segmentation training datasets were augmented through random vertical/horizontal flips, rotations and transpositions, each applied with a probability of 0.5.

**BUSI dataset.** BUSI dataset (Al-Dhabyani et al., 2020) is a publicly available dataset for breast tumor segmentation in ultrasound imaging. It consists of 780 grayscale breast ultrasound images in PNG format, collected from 600 female patients (aged 25–75) and categorized into three classes: 133 images with no visible tumors (normal), 437 images of non-cancerous lesions (benign) and 210 images of confirmed cancerous tumors (malignant). We utilized 647 benign and malignant images, resized to $256 \times 256$.

**GlaS dataset.** GlaS dataset (Valanarasu et al., 2021) is a widely used dataset, specifically designed for gland segmentation. It contains 165 Hematoxylin and Eosin (H&E) stained histology RGB images. Our study utilized 165 images, resized to $512 \times 512$. Despite the predefined train-test division, we re-partitioned them into training (80%) and testing (20%) subsets using a randomized split with the seed 42. Such approach ensures a fair comparison with competitors and the way we split BUSI and CVC datasets.

Table 1: Performance comparison of state-of-the-art segmentation models across three clinically distinct medical imaging scenarios. Results report average intersection over union (IoU) and F1 scores with standard deviation over three random runs

| Methods | BUSI | | GlaS | | CVC | |
|---|---|---|---|---|---|---|
| | IoU ↑ | F1 ↑ | IoU ↑ | F1 ↑ | IoU ↑ | F1 ↑ |
| U-Net (Ronneberger et al., 2015) | 57.22±4.74 | 71.91±3.54 | 86.66±0.91 | 92.79±0.56 | 83.79±0.77 | 91.06±0.47 |
| Att-Unet (Oktay et al., 2018) | 55.18±3.61 | 70.22±2.88 | 86.84±1.19 | 92.89±0.65 | 84.52±0.51 | 91.46±0.25 |
| U-Net++ (Zhou et al., 2018) | 57.41±4.77 | 72.11±3.90 | 87.07±0.76 | 92.96±0.44 | 84.61±1.47 | 91.53±0.88 |
| U-NeXt (Valanarasu & Patel, 2022) | 59.06±1.03 | 73.08±1.32 | 84.51±0.37 | 91.55±0.23 | 74.83±0.24 | 85.36±0.17 |
| Rolling-UNet (Liu et al., 2024a) | 61.00±0.64 | 74.67±1.24 | 86.42±0.96 | 92.63±0.62 | 82.87±1.42 | 90.48±0.83 |
| U-Mamba (Ma et al., 2024) | 61.81±3.24 | 75.55±3.01 | 87.01±0.39 | 93.02±0.24 | 84.79±0.58 | 91.63±0.39 |
| UKAGNet (Drokin, 2024) | 63.45 | **77.64** | 87.31 | 93.23 | 76.85 | 86.91 |
| U-KAN (Li et al., 2025) | 63.38±2.83 | 76.40±2.90 | 87.64±0.32 | 93.37±0.16 | 85.05±0.53 | **91.88±0.29** |
| U-FunKAN (Ours) | **68.49±0.62** | 77.37±0.58 | **88.02±0.24** | **93.50±0.12** | **85.93±0.72** | 91.42±0.61 |

Table 2: Efficiency comparison of floating-point operations (Gflops) and trainable parameters (Params) across state-of-the-art segmentation algorithms. Results are measured for $256 \times 256$ input resolution using THOP Python tool for profiling PyTorch models

| Methods | Gflops ↓ | Params (M) ↓ |
|---|---|---|
| U-Net (Ronneberger et al., 2015) | 524.2 | 34.53 |
| Att-Unet (Oktay et al., 2018) | 533.1 | 34.9 |
| U-Net++ (Zhou et al., 2018) | 1109 | 36.6 |
| U-NeXt (Valanarasu & Patel, 2022) | 4.58 | **1.47** |
| Rolling-UNet (Liu et al., 2024a) | 16.82 | 1.78 |
| U-Mamba (Ma et al., 2024) | 2087 | 86.3 |
| U-KAN (Li et al., 2025) | 14.02 | 6.35 |
| U-FunKAN (Ours) | **4.35** | 3.6 |

**CVC-ClinicDB dataset.** CVC-ClinicDB dataset (Bernal et al., 2015) is a dataset for polyp segmentation in colonoscopy images. It contains 612 high-resolution colonoscopy RGB frames, extracted from 29 video sequences with varied lighting conditions, specular reflections and mucosal textures. All images were resized to $256 \times 256$.

We measured anatomical structures segmentation accuracy on BUSI, GlaS and CVC-ClinicDB datasets in terms of IoU and F1 scores. To ensure full reproducibility, we released CSV files specifying the data splits for each dataset in our GitHub repository.

Table 1 presents a quantitative evaluation of our proposed U-FunKAN model against several leading segmentation architectures, including convolutional U-shaped networks, attention-based variants and recent hybrid models, incorporating Mamba and KAN layers. The reported metrics are averaged over the last fifty epochs, coupled with uncertainty estimations, derived from three independent training runs with the seeds: 50, 100 and 150. The analysis reveals that U-FunKAN sets a new state-of-the-art, achieving the highest IoU on all three datasets, also being the most efficient algorithm in terms of Gflops (see Table 2). U-FunKAN also attains the highest F1-score on GlaS datasets, while minor underperforming on BUSI and CVC datasets compared to UKAGNet (Drokin, 2024) and U-KAN (Li et al., 2025), respectively, which requires $\times 3$ greater computational complexity (in Gflops) and $\times 1.7$ more parameters than our approach. The observed performance profile, characterized by a high IoU and a slightly lower F1-score, suggests that the model produces fewer false positives (a characteristic highly prioritized by IoU) at the cost of a modest increase in false negatives (as reflected by the F1-score). Deeper U-FunKAN architectures mitigate this trade-off, achieving state-of-the-art quality in both metrics (see Table 3). Beyond accuracy, U-FunKAN exhibits a low variance across multiple runs, indicating training stability and reliability.

The comparative analysis of U-FunKAN model variants with different channel settings is reported in Table 3. The proposed channel setting: 32 $(C_1) \rightarrow$ 64 $(C_2) \rightarrow$ 128 $(C_3)$, – achieves an optimal balance between computational efficiency and segmentation performance, yielding state-of-the-art results. Deeper configurations can further improve accuracy, while they incur additional computational overhead.

Table 3: Ablation study on impact of the channel scaling in U-FunKAN on segmentation performance and efficiency. The best intersection over union (IoU) and F1 scores are reported

| U-FunKAN | | | BUSI | | | |
|---|---|---|---|---|---|---|
| $C_1$ | $C_2$ | $C_3$ | IoU ↑ | F1 ↑ | Gflops ↓ | Params (M) ↓ |
| **32** | **64** | **128** | 69.11 | 77.95 | **4.35** | **3.6** |
| 64 | 96 | 128 | 69.94 | 78.42 | 10.84 | 4.1 |
| 128 | 160 | 256 | 69.49 | 78.39 | 40.42 | 15.7 |
| 256 | 320 | 512 | **70.62** | **79.31** | 161.43 | 62.4 |

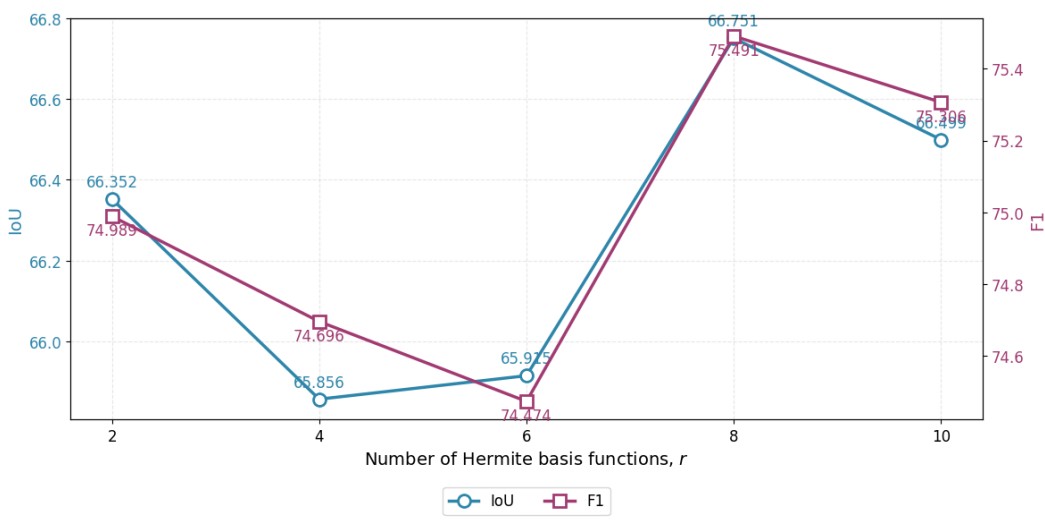

Figure 4: Ablation study on impact of the number of the basis Hermite functions in U-FunKAN on breast cancer segmentation quality on BUSI ultrasound dataset. Averaged IoU and F1 scores are reported over the last fifty epochs for each experiment.

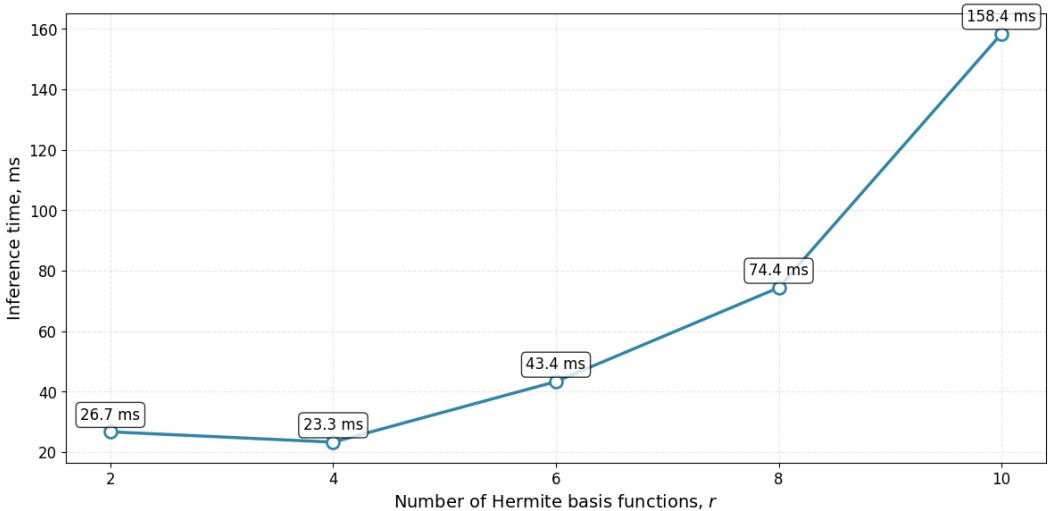

Figure 5: Ablation study on impact of the number of the basis Hermite functions in U-FunKAN on efficiency. Median inference times are reported over 100 runs for a $256 \times 256$ input on Intel Core i7-14700HX.

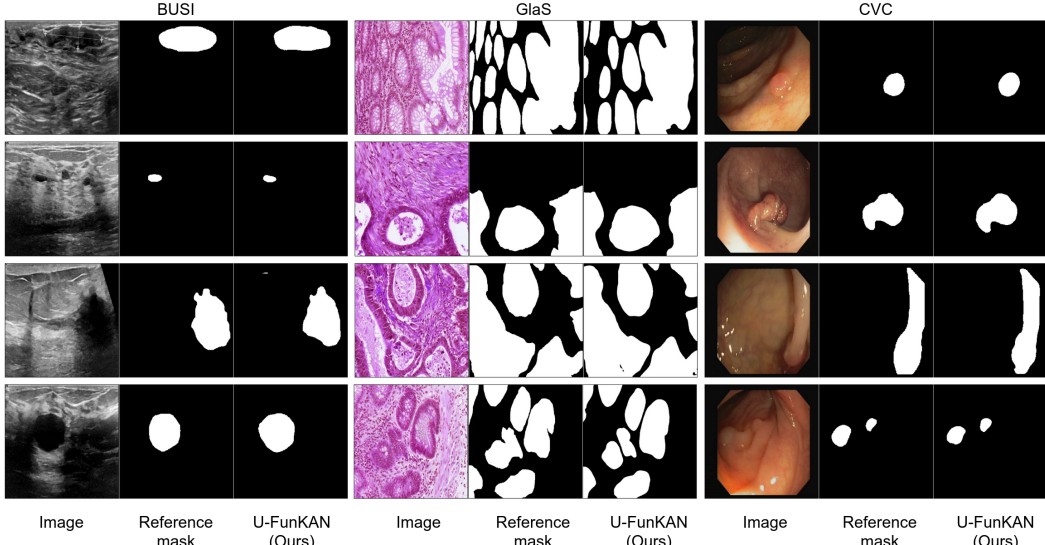

Figure 6: Qualitative results of U-FunKAN over three heterogeneous medical scenarios.

Fig. 4 shows an ablation study evaluating the sensitivity of U-FunKAN's segmentation quality to the number of Hermite basis functions, $r$. Models were trained from scratch with learning rate $10^{-4}$ till convergence on BUSI dataset to ensure a fair comparison. The results demonstrate that both IoU and F1-score improve as $r$ increases from the default value of 6 to 8 or 10. However, this extra gain in accuracy comes at the cost of increased computational latency, as detailed in Fig. 5.

Fig. 6 presents qualitative segmentation results of the proposed U-FunKAN model across three heterogeneous medical imaging scenarios: breast cancer detection in ultrasound images, glands segmentation in histopathology and polyps segmentation in colonoscopy images.

Furthermore, to demonstrate that our method has general applicability beyond binary segmentation, we apply FunKAN to the task of MRI reconstruction, outperforming KAN learnable backbones (Seydi, 2024) and classical state-of-the-art Kellner method (Kellner et al., 2016) for Gibbs-ringing suppression. In particular, we leverage FunKAN as a backbone architecture for MRI enhancement with the following setup:

1. Embedding: $5 \times 5$ convolution, projecting an input image into 16-dimensional feature space.

2. Lifting: $3 \times 3$ convolution, projecting ReLU pre-activated features into 32-dimensional feature space, maintaining the spatial resolution.

3. Backbone: sequence of three FunKAN blocks, interconnected with skip connections in 32-dimensional feature space ($n = 32$), encapsulating the spectral encoding of each inner function over the first six Hermite basis functions ($r = 6$).

4. Projection: $3 \times 3$ convolution, projecting ReLU pre-activated features into 16-dimensional feature space, maintaining the spatial resolution.

5. Restoration: $1 \times 1$ convolution, projecting ReLU pre-activated features to the target color space.

This model is trained in a supervised manner using mean squared error loss function on IXI dataset, mitigating Gibbs-ringing after discarding 75% of high-frequency information:

$$\mathcal{L}_{enh} = \frac{1}{N} \sum_{i=1}^{N} ||I_i^* - I_i^1||_2^2, \tag{10}$$

where $N$ – batch size, equals to 8.

| Methods | IXI | |
| --- | --- | --- |
| | PSNR ↑ | TV |
| $I^0$ | 31.33 | 1476.55 |
| $I^1$ | – | 1255.40 |
| Kellner (Kellner et al., 2016) | 31.09 | 1120.05 |
| MLP (Cybenko, 1989) | 37.96 | 1145.57 |
| KAN (Liu et al., 2024b) | 38.10 | 1161.63 |
| ChebyKAN (SS et al., 2024) | 38.01 | 1156.56 |
| HermiteKAN (Seydi, 2024) | 38.04 | 1161.31 |
| FunKAN (Ours) | **39.05** | **1174.86** |

Table 4: Application of FunKAN as MRI enhancement backbone within the convolutional architecture. Results report average peak signal-to-noise ratio (PSNR) and total variation (TV) across 2617 test images from IXI dataset. $I^0$, $I^1$ denote artifact-corrupted and artifact-free images, respectively.

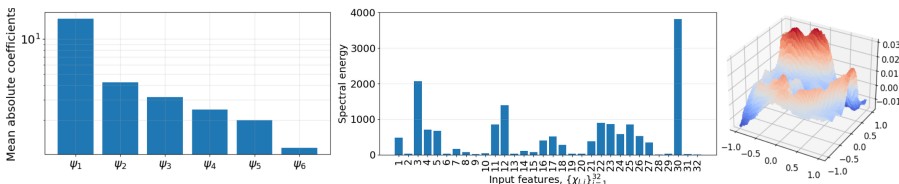

Figure 7: Interpretability of FunKAN as MRI enhancement backbone. Aggregated contribution of Hermite functions $\{\psi_k\}_{k=1}^r$ across all learned inner functions. Feature ranking intrinsics, computed as the total spectral energy, towards feature selection. A learned inner function $\varphi_{l,i} \in \mathbb{R}^{h \times w}$.

**IXI dataset** comprises approximately 600 MRI scans ($255 \times 255$) from healthy subjects, collected using 1.5T and 3T scanners from Philips and GE Healthcare. It is composed of T1, T2 and PD 3D volumes, encoded in NIFTI format. Firstly, the intersection of these volumes was taken, producing 577 volumes that have all three modalities: T1, T2 and PD. Then, the first 400 volumes were utilized to synthesize the training set, the next 100 volumes to create the testing set and the rest of the data to generate the validating set, forming 10473, 2617 and 2016 image pairs, respectively.

A quantitative comparison (see Table 4) of backbones within the same convolutional architecture for MRI enhancement on IXI dataset demonstrates that substituting MLP with KAN and subsequently retraining the entire model results in a modest PSNR improvement of 0.1 dB. In contrast, FunKAN backbone achieves a markedly higher gain, outperforming all considering here KANs (B-spline, Chebyshev, Hermite) by 1 dB, owing to its inherent capacity to incorporate the geometric relations of visual data by design. Performance is benchmarked with superior restoration quality indicated by higher PSNR values and TV measures approaching the reference $I^1$.

From an interpretability standpoint (see Fig. 7), the Hermite decomposition of a feature provides a spectral characterization of its complexity by construction. A concentration of spectral energy in the low-order Hermite coefficients indicates a smooth inner function, which is empirically associated with robust, generalizable features. Conversely, significant energy in higher-order coefficients suggests a propensity for overfitting and heightened adversarial vulnerability, as high-frequency components often correspond to non-robust, data-specific artifacts (Harder et al., 2021).

## 5 CONCLUSION

This paper introduced FunKAN, a novel neural framework that advances the state-of-the-art in medical image segmentation. The core of our contribution is a generalization of the Kolmogorov-Arnold representation theorem to functional spaces, which we have empirically validated. Through extensive evaluations on diverse tasks, including breast tumor detection in ultrasound, gland segmentation in histology and polyp identification in colonoscopy, the proposed U-FunKAN architecture demonstrated superior performance over contemporary models, such as convolutional networks, attention-based mechanisms and recent architectures based on Mamba and KAN. To support reproducibility and future research, our code will be made publicly available upon acceptance of this paper.

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
