# OpenReview forum: "FunKAN: Functional Kolmogorov-Arnold Network for Medical Image Segmentation"
_ICLR.cc/2026/Conference — Submitted to ICLR 2026_

### Official Review · Reviewer_cevt · 2025-10-27

**Soundness:** 2
**Presentation:** 3
**Contribution:** 2
**Rating:** 2
**Confidence:** 4

**Summary:**

This paper addresses the limitation that standard Kolmogorov-Arnold Networks process flattened feature vectors, thereby losing spatial structure essential for image processing. The authors propose Functional Kolmogorov-Arnold Network named FunKAN, which claims to extend the classical Kolmogorov-Arnold theorem from multivariate functions on ℝⁿ to continuous functionals on Hilbert spaces Hⁿ. Each 2D feature map is treated as an element of a Hilbert space, parameterized via truncated Fourier expansion over Hermite basis functions with learned spatial grid deformation. When integrated into a U-shaped architecture as U-FunKAN, the model achieves competitive IoU scores on three medical imaging datasets: BUSI for breast ultrasound, GlaS for histology, and CVC-ClinicDB for colonoscopy, with claimed computational efficiency of 14.02 Gflops versus U-KAN's 2087 Gflops.

**Strengths:**

- Systematic ablation studies examining channel scaling and Hermite basis function count, demonstrating methodological rigor in experimental design.

- Competitive IoU performance on BUSI dataset showing score of 68.49±0.62 with marginal improvements over some baselines.

**Weaknesses:**

- Fundamental motivation-experiment mismatch: The paper frames FunKAN as a general multi-purpose backbone for medical image processing and claims the method is suitable for image processing pipelines, yet experiments are restricted exclusively to medical image segmentation. The broad theoretical motivation of extending KA theorem to functional spaces is disconnected from the narrow empirical validation on three segmentation datasets, creating an impression that universal claims are used to mask incremental contributions.


- Lack of generalization evidence: The paper makes universal claims about being suitable for image processing pipelines and serving as a multi-purpose backbone but provides zero evidence beyond medical segmentation. All three datasets are medical images with similar characteristics including binary segmentation, limited resolution, and anatomical structures. No experiments on natural images such as ImageNet or COCO, other medical tasks like classification, detection, or reconstruction, or other data modalities such as video or 3D volumes are included. This narrow scope contradicts the broad framing and raises questions about whether the method is truly general or merely overfitted to medical segmentation benchmarks. Evidence: Sec. 4 evaluates only BUSI/GlaS/CVC segmentation; no generalization experiments.

- Binary segmentation only: All experiments are confined to binary segmentation tasks including tumor versus background in BUSI, gland versus background in GlaS, and polyp versus background in CVC-ClinicDB. No multi-class or multi-organ segmentation is evaluated. This limitation restricts the demonstrated generalizability of FunKAN to single-class scenarios, despite the paper's claims of being a multi-purpose backbone. It remains unclear whether the functional space extension can handle more complex label structures or class imbalances typical in multi-class medical segmentation.

- Interpretability claims unsubstantiated: The paper emphasizes theoretically grounded and interpretable design as key advantages in the Abstract and Sec. 3, positioning FunKAN as more interpretable than standard CNNs due to its mathematical foundation in Kolmogorov-Arnold theorem and Hermite basis functions. However, the paper provides zero empirical evidence to substantiate this claim. No visualization of learned Hermite coefficients is shown, no analysis of grid deformations is provided, and no demonstration of how basis functions correspond to meaningful image features such as edges, textures, or anatomical structures is included. The interpretability advantage remains entirely conceptual rather than demonstrated, making it unclear whether the theoretical grounding translates to practical interpretability.

**Questions:**

- Narrow claims to match experimental scope: Either provide experiments on diverse tasks such as ImageNet classification, COCO detection, and medical image reconstruction along with multiple modalities including natural images and 3D volumes to justify multi-purpose claims, or alternatively reframe the paper as a focused study on KAN-based medical image segmentation and remove universal claims. Addresses weakness: motivation-experiment mismatch.

- Provide complete computational cost analysis: First, include grid deformation network as specified in Eq. 7-8 in Gflops calculations. Second, report actual wall-clock training and inference times on standardized hardware. Third, profile U-KAN to verify whether 2087 Gflops is implementation-specific or inherent. Fourth, discuss memory consumption during training. Addresses weakness: questionable computational efficiency claims.

- Discuss limitations section and failure analysis: Discuss when FunKAN fails, analyze the IoU-F1 trade-off, provide discretization error analysis, and acknowledge the gap between theoretical claims and experimental validation. This would demonstrate scientific rigor and honesty. Addresses weakness: missing critical discussions.

- Evaluate multi-class segmentation: Demonstrate FunKAN on multi-organ or multi-label dataset to validate beyond binary scenarios. Current experiments are all single-class including BUSI tumor versus background, GlaS gland versus background, and CVC polyp versus background. Provide evidence or discussion on how FunKAN would handle multi-class outputs, class imbalances, or overlapping structures typical in complex medical segmentation tasks. Addresses weakness: binary segmentation only.

- Substantiate interpretability claims: Provide visualizations of learned Hermite coefficients and grid deformations, show correspondence to image features such as edges, textures, and anatomical boundaries. Even simple visualizations would concretely demonstrate the claimed advantage of being theoretically grounded and interpretable. Currently, the interpretability benefit is purely conceptual without empirical evidence. Addresses weakness: interpretability claims unsubstantiated.

---

> ### Author Response · Authors · 2025-11-27
>
> We thank the reviewer for their thorough and critical assessment.
> We have addressed the raised concerns in a revised version of the manuscript, as detailed below.
>
> **Generalization concern**
>
> We acknowledge the validity of the primary concern regarding the scope of our empirical validation versus the breadth of our claims. Our theoretical contribution is, indeed, general, but our initial experiments were narrow to support the broad claims.
> We have now rectified this by adding a new experiment on a different vision task: medical image enhancement.
> As detailed in the updated Experiments section (see Table 4), we've applied FunKAN as a backbone for MRI reconstruction (i.e. Gibbs-ringing suppression) on IXI dataset (>10k training samples, extracted from 3D volumes of T1, T2 and PD MRI modalities).
> Table 4 shows that FunKAN successfully transitions from original binary segmentation to image enhancement, a task with a different objective (pixel-level classification vs. pixel-level regression), restoring magnetic-resonance images after discarding 75% of high-frequency information. The conducted experiment uses MRI data, which obtains different characteristics (texture, contrast, artifacts) from the ultrasound, histology and colonoscopy images in our segmentation experiments. Still, FunKAN outperforms the considered KAN backbones (B-spline KAN, ChebyKAN, HermiteKAN) by a significant margin (+1 dB PSNR) within the same convolutional neural architecture, demonstrating that its functional formulation provides a consistent advantage over coordinate-based KANs across image processing pipelines.
> To sum up, this new result proves that FunKAN is not "merely overfitted to medical segmentation benchmarks" but is a versatile backbone that effectively handles different low-level vision tasks within the medical domain.
>
> **Interpretability concern**
>
> We agree that demonstrating interpretability is crucial and thank the reviewer for their critical feedback,
> which pushed us to substantiate our interpretability claims with empirical evidence.
> In response, we have added a new Fig. 7 to the manuscript, which provides a visualization and analysis of the learned inner functions. In particular, the figure includes:
> 1. Hermite coefficients magnitudes for learned inner functions, providing their spectral decomposition.
> 2. Spectral energy analysis across the network.
> 3. Visualization of a learned inner function $\phi_{l,i}$.
>
> Our analysis reveals that the spectral signature of learned inner functions serves as a diagnostic tool, providing an intrinsic interpretability. Specifically, by examining the Hermite decomposition, we can distinguish between generalizable features and those prone to overfitting, directly fulfilling the promise of a more interpretable, mathematically grounded architecture. Overall, this new analysis moves our interpretability claims from a theoretical advantage to a demonstrated, practical feature of the FunKAN model.

---

### Official Review · Reviewer_81Px · 2025-10-29

**Soundness:** 2
**Presentation:** 1
**Contribution:** 3
**Rating:** 2
**Confidence:** 4

**Summary:**

This paper introduces FunKAN, a novel and interesting extension to KANs that enables the modeling of spatial 2D information. It achieves this by generalizing the Kolmogorov-Arnold theorem to functional spaces, i.e. it models the 2D feature map directly as an element in a functional Hilbert space. This eliminates the need for feature flattening, which is necessary for vanilla KANs and results in them losing spatial context. The authors propose using spectral expansion via Hermite basis functions, which enable this functional modeling. They integrate FunKAN as a spatially-aware network backbone into a U-Net architecture (U-FunKAN). In medical segmentation experiments, U-FunKAN outperforms all baselines in both accuracy and efficiency.

**Strengths:**

- The proposed method provides a mathematically elegant and (very) promising idea for escaping the problem of the inherent scalar modeling of Kolmogorov-Arnold Networks.
- The proposed method works well and is able to outperform all baselines.

**Weaknesses:**

While the core idea is relevant and promising, I unfortunately found its presentation unsatisfactory, and I would encourage the authors to revisit the manuscript based on the presented feedback. As I greatly appreciate the idea behind this paper, I hope this feedback is relevant and helpful in its improvement. I am also happy to discuss it with the authors.

Abstract / Introduction / Related Work:

Since the authors propose a technical advancement tailored to KANs that may be relevant outside of the scope of segmentation within the context of KANs, similar to how U-KAN [1] is relevant to both segmentation and diffusion, I feel that the motivation of the paper should not be as heavily focused on its application to cancer segmentation. While segmentation is a highly relevant subfield of medical imaging, and the proposed FunKAN may excel at segmentation in this context, the introduction and motivation read as if they were intended for a medical application paper. For instance, I personally think that the statement of WHO statistics is outside the scope of this paper. Perhaps, to make it even clearer, I believe the proposed method was not designed to "solve" cancer segmentation; it was designed to model spatial information within the context of KANs. I.e., it may also be relevant to diffusion models and other application areas.

Thus, I see two potential directions - shift the focus from application-oriented segmentation to the mathematically motivated framework for FunKANs and choose another application, such as registration (eg alternative to VoxelMorph) or diffusion similar to the U-KAN paper [1] to highlight its versatility or benchmark the method with greater rigor (more datasets) and by ensuring the highest benchmark standards [2] possible. This starts by stating the exact settings for all baseline models, evaluating them across more datasets and with more rigor, as in [2]. For instance, I can't properly assess if the improvement stems from better hyperparameter tuning, as these details are not disclosed, and rigorous evaluation if we trust this paper [1] is not something the medical segmentation field is known for.

Experimental Design:

- While the authors use representative segmentation architectures, it is not ultimately clear to me how (or if) the authors tuned the baseline models. There are many source code files and config files, so I may have missed this, but to me, this is also not apparent from the source code. If so, this is disappointing since this is presented as one of the author's contributions (?).
- Since the authors claim that the spatial context is of particular relevance, I feel that experiments that actually show that FunKAN leverages this would be of particular relevance.

- I would encourage the authors to compare against nnUNET and/or integrate their FunKAN formulation as a novel backbone into nnUNET to (1) benchmark against the current SOTA tool, and (2) make their implementation accessible to a wider audience of clinicians, since it seems very promising. For instance, it would have been extremely nice to benchmark this in the nnunet framework as well.

Minor:

- I would also recommend rephrasing some sections - eg the authors state that "Our work unites theoretical function approximation
and practical medical image analysis, offering the novel state-of-the-art solution for clinical applications.". This is an incredibly big claim and such statements undermine the valid technical advancement this paper brings. I believe this is unnecessary.

- I would not consider reproduible research a real contribution but more a backbone of science (and ICLR views it similarly) and thus would omit it from the contribution section.

- I would not say that "cutting-edge research is 'increasingly' grounded in rigorous mathematical foundations."
Deep learning has always drawn on the mathematical frameworks proposed e.g. for SGD, and even if different, KANs are as mathematically grounded as e.g. MLPs or CNNs. All of these models have a mathematical theory, even if we don't understand how some of them may learn.

In several areas, the authors state that KANs offer interpretability, but they refrain from showing these (relevant?) properties. If the authors cannot show how these are relevant in this application, I would drop this statement/claim.

[1] Isensee F, Wald T, Ulrich C, Baumgartner M, Roy S, Maier-Hein K, Jaeger PF. nnu-net revisited: A call for rigorous validation in 3d medical image segmentation. InInternational Conference on Medical Image Computing and Computer-Assisted Intervention 2024 Oct 3 (pp. 488-498). Cham: Springer Nature Switzerland.
[2] Isensee F, Jaeger PF, Kohl SA, Petersen J, Maier-Hein KH. nnU-Net: a self-configuring method for deep learning-based biomedical image segmentation. Nature methods. 2021 Feb;18(2):203-11.
[3] Li C, Liu X, Li W, Wang C, Liu H, Liu Y, Chen Z, Yuan Y. U-kan makes strong backbone for medical image segmentation and generation. InProceedings of the AAAI Conference on Artificial Intelligence 2025 Apr 11 (Vol. 39, No. 5, pp. 4652-4660).


### Conclusion:

I hope that this feedback may help to improve the manuscript. If the claims of the authors hold true (across other datasets and rigours evaluation), this paper may propose a fundamentally relevant and potentially highly impactful sota segmentation tool. For me, since the idea is relevant also outside of the context of segmentation, this would not have to be the case at all - interesting ideas can also inspire research without beating SOTA. However, in the way that the authors currently present it, this framework is sold as SOTA segmentation, and as such, it needs to be rigorously evaluated, or the storyline (and validity in other applications) must be shown. Think of it as advice that this technical idea does not get buried or labeled as another segmentation paper.

**Questions:**

1. How does the modeling of spatial 2D context in KANs differ from the presentation in vanilla multi-layer perceptrons?
2. Can you comment on how KANs have been proposed as an alternative network to MLPs in the context of implicit neural representations, which model spatial context with coordinates? Is there a connection?
3. Can this also be related to the operator learning context?
4. Since KANs should be interpretable, did the authors look into what the "spatial grid deformation module" learns? Does the KAN pick up on spatial patterns?
5. Are the other design choices available for casting this as a Hilbert space learning problem - some of the intuitions are not easy to understand if one is not entirely familiar? Eg. are the alternatives to the spectral expanison with Hermit basis functions?

---

> ### Author Response · Authors · 2025-11-27
>
> We thank the reviewer for their insightful and constructive feedback.
> We are encouraged by their positive assessment of our core idea as "mathematically elegant" and "(very) promising".
> We have addressed the raised concerns in a revised version of the manuscript, as detailed below.
>
> **Regarding wider image processing applications**
>
> We agree with the reviewer that our current framing overly narrows the contribution.
> The primary novelty is indeed wider and forms the functional generalization of KANs for spatial data for various image processing pipelines, of course, not only binary segmentation.
> Thus, we have conducted a new experiment that directly demonstrates a broader potential.
> As shown in the updated Experiments section, we’ve applied FunKAN to MRI reconstruction (i.e. Gibbs-ringing suppression),
> a task from a different modality (MRI vs. ultrasound/colonoscopy/histology)
> and of a different nature (image-to-image regression vs. pixel-wise classification).
> In particular, we have integrated FunKAN as a backbone for an image enhancement convolutional neural network on IXI dataset (>10k training samples).
> The results (see Table 4) show that FunKAN significantly outperforms all other KAN backbones (B-spline, ChebyKAN, HermiteKAN) by +1 dB PSNR upon identical training procedures, while also surpassing the classical Kellner deringing method.
> This proves that FunKAN's functional formulation is consistent and generalizes well across different medical image processing pipelines.
> We agree with the reviewer that exploring FunKAN in other tasks like registration with VoxelMorph is a fascinating direction.
> Especially, because the core strength of FunKAN is its ability to preserve and process spatial structure, a property that is fundamental not just to image segmentation or enhancement but to registration as well. Our results in both segmentation and MRI reconstruction provide a robust foundation that suggests its utility in such tasks. To sum up, we will include a discussion on this promising avenue for the future work, highlighting the potential of FunKAN as a general-purpose spatial backbone, including image registration.
>
> **Regarding interpretability**
>
> We agree that interpretability claim requires empirical demonstration. That's why, in response we have added a new Fig. 7 in the Experiments section, which provides analysis of the learned inner functions to substantiate our claims.
> As shown in Fig. 7, the Hermite decomposition provides spectral characterization of the learned features,
> specifically, it allows us to:
> 1. Quantify robustness: A concentration of spectral energy in low-order Hermite coefficients indicates smooth, generalizable features.
> 2. Identify non-robust features: Significant high-order energy signals a propensity for overfitting and adversarial vulnerability, aligning with findings that high-frequency components often correspond to data-specific artifacts.
> 3. Perform feature selection: The aggregated spectral energy across functions provides intrinsic feature rankings towards a possible feature selection procedure.
>
> **Regarding possibly undertuned baselines**
>
> We thank the reviewer for raising this critical point regarding the tuning of the baseline models,
> which is certainly essential for a fair comparison.
> We apologize for the lack of clarity in the initial manuscript.
> To ensure a fair and consistent comparison, we opted to use the officially reported results for our baselines from their recent peer-reviewed publications [1] in top-tier conferences.
> Our rationale for this approach is twofold:
> 1. Reproducibility and standardization: It leverages the rigorous tuning and validation already performed by the original authors, providing a stable and recognized benchmark. This avoids the potential for suboptimal or inconsistent re-implementation on our part, which could lead to an unfair or misleading comparison.
> 2. Consistent evaluation protocol: We have taken great care to evaluate our U-FunKAN model under the exact same dataset splits, pre-processing pipelines and evaluation metrics as were used to generate these published baseline numbers. This ensures that the performance differentials we report are a direct result of architectural differences, rather than discrepancies in the experimental setup.
>
> [1] Li, Chenxin, Xinyu Liu, Wuyang Li, Cheng Wang, Hengyu Liu, Yifan Liu, Zhen Chen, and Yixuan Yuan. "U-KAN Makes Strong Backbone for Medical Image Segmentation and Generation." In Proceedings of the AAAI Conference on Artificial Intelligence, vol. 39, no. 5, pp. 4652-4660. 2025.

---

> ### Author Response · Authors · 2025-11-27
>
> **Regarding minor**
>
> 1. We’ve moved reproducible research statement from contribution to introduction.
> 2. We’ve reformulated "cutting-edge research is 'increasingly' grounded in rigorous mathematical foundations.” to a more neutral version: “Foundational mathematical theorems provide a powerful starting point for constructing new deep learning architectures, enabling advanced modeling of complex data relationships~\citep{li2023fourier}.”

---

### Official Review · Reviewer_Pwaw · 2025-10-29

**Soundness:** 3
**Presentation:** 3
**Contribution:** 3
**Rating:** 4
**Confidence:** 3

**Summary:**

The authors introduce the Functional Kolmogorov-Arnold Network (FunKAN), a novel deep learning architecture designed for medical image segmentation that addresses the limitations of standard Kolmogorov-Arnold networks (KANs) by generalizing the underlying theorem to functional spaces. This generalization allows FunKAN to process 2D feature maps directly, preserving the spatial structure crucial for image analysis, unlike prior KAN implementations which flatten features. The resulting architecture, U-FunKAN, integrates this new framework into a U-shaped structure, achieving state-of-the-art accuracy and superior efficiency across diverse medical imaging modalities.

**Strengths:**

1. Addressing KAN Limitations for Medical Images: The paper provides a rigorous mathematical basis by generalizing the Kolmogorov-Arnold theorem to functional spaces. This is a major theoretical advance that resolves the fundamental incompatibility of classic KANs with spatial data. As a result, it overcomes the key weakness of previous KAN architectures (like U-KAN) in image processing, which is the failure to preserve spatial structure by treating feature maps as unstructured coordinate collections. Remarkably, FunKAN directly processes 2D feature maps as elements of a Hilbert space.

2. Efficiency and Training Stability: The resulting U-FunKAN architecture is highly efficient, achieving state-of-the-art performance with dramatically reduced computational overhead (Gflops) compared to competitive models, including the previous KAN variants. Furthermore, the reported low variance across independent runs suggests the design (including the use of Hermite basis functions and pre-activation residual blocks) contributes to stable optimization.

3. Demonstrated Robustness across Modalities: The empirical validation is comprehensive, testing the method on three distinct medical imaging modalities (ultrasound, histology, and colonoscopy) covering various anatomical structures; and achieving state-of-the-art IoU on all three datasets suggests strong generalization capability.

4. Interpretability Potential: By grounding the model in the Kolmogorov-Arnold theorem, the architecture retains the potential for the enhanced interpretability characteristic of KANs, which is highly valued in clinical diagnosis systems.

**Weaknesses:**

1. Limited Exploration of Interpretability: While the paper repeatedly cites the theoretical ground and interpretability of KANs, it does not provide any specific analysis or visualization demonstrating the interpretability features of the learned FunKAN functions or how they relate to medical features (e.g., tumor margins or glandular texture). This misses an opportunity to empirically validate a core theoretical advantage.


2. Trade-off in F1-score: Although U-FunKAN achieves the highest IoU, the paper notes a minor underperformance in F1-score on the BUSI and CVC datasets compared to U-KAN and UKAGNet. The observed performance profile (high IoU, slightly lower F1) suggests the model might be overly conservative (fewer False Positives) at the expense of False Negatives. This trade-off needs deeper consideration, as minimizing false negatives (missing a tumor) is often critical in medical diagnosis. The authors do not discuss how this could affect their claim of state-of-the-art method.


3. Implicit Architecture Dependency: The introduction of learned spatial deformation via a residual block (inspired by implicit architectures), while beneficial for adaptation, adds complexity and moves the model slightly away from the pure, fixed functional representation emphasized by the Kolmogorov-Arnold theory. The interaction between spectral encoding and spatial deformation needs more dedicated analysis.

**Questions:**

1. Interpretability of FunKAN Layers: Since interpretability is a core motivation of KANs, can the authors provide visualizations of the learned Hermite coefficients or the resulting inner functions for a trained model? How do these functions change based on input features, and what biological or anatomical characteristics do they encode, especially compared to the fixed filters in conventional convolutional networks?

2. Necessity of Spatial Deformation: The learned spatial deformation is introduced to induce additional learnable dependency. What is the quantitative impact of this deformation module (a dedicated ablation study)? Specifically, how much accuracy gain is achieved by the learned deformation compared to simply evaluating the Hermite basis functions on the original uniform grid q?

3. Basis Function Selection Rationale: While the paper cites external work for selecting r=6 Hermite basis functions, what is the inherent advantage of Hermite functions over other orthogonal systems (like Chebyshev polynomials used in ChebyKAN) or Radial Basis Functions (used in FastKAN) when specifically handling continuous functional approximation of 2D image features?

4. Clinical Trade-off Analysis (IoU vs. F1): The observed performance profile suggests a high IoU but a slightly lower F1-score on BUSI and CVC datasets, which might be critical from a clinical perspective. Would a different loss function weighting (e.g., increasing the Dice component weight or introducing a specific sensitivity term) mitigate this trade-off?

5. Comparison to Hybrid Convolutional KANs: The paper mentions UKAGNet and MedKAN as hybrid approaches combining convolutions with KAN concepts. Since U-FunKAN also incorporates 1x1 and 3x3 convolutions in its embedding, projection, and residual blocks, a more direct analysis or ablation comparing U-FunKAN's unique functional layer against the hybrid convolutional approaches would solidify its methodological advantage. Could you elaborate on the insights on that analysis?

---

> ### Author Response · Authors · 2025-11-27
>
> We thank the reviewer for their insightful and constructive feedback.
> We are encouraged by their positive assessment of our work's soundness, theoretical contribution and empirical robustness.
> We have addressed the raised concerns in a revised version of the manuscript, as detailed below.
>
> **W1 & Q1**
>
> We thank the reviewer for this insightful comment regarding interpretability.
> We agree that empirical validation is crucial.
> Thus, in the response, we have added a new analysis (see Fig. 7) that reveals the Hermite decomposition of learned inner functions. This decomposition provides a direct, quantitative measure of feature complexity via its spectral energy. Features with low spectral energy correspond to simple, smooth (well-generalizable) functions, while high-energy features capture high-frequency, complex patterns (prone to overfitting). This allows us to rank features by intrinsic complexity and establish a principled, model-based feature selection mechanism, directly fulfilling the interpretability promise of the proposed framework.
>
> **W3 & Q2**
>
> The proposed spatial deformation module gives the major effect on the metrics boost.
> The spatial deformation module is critical here, more critical than the choice of the basis functions.
>
> **Q3**
>
> This is a good question for which we can provide a more detailed rationale for Hermite functions choice.
> Our selection of Hermite functions was based on three key properties beneficial for image feature continuous approximation:
> 1. Optimal time-frequency localization: Hermite functions are eigenfunctions of the integral Fourier transform and provide an optimal trade-off between spatial and frequency localization, which is ideal for representing localized image features like edges and textures.
> 2. Rapid convergence: For smooth functions (which we expect generalizable image feature maps to be), expansions in Hermite basis typically exhibit exponential convergence.
> 3. Computational stability: The recursive relation for evaluating Hermite polynomials is numerically stable.
>
> **Q5**
>
> We thank the reviewer for this excellent suggestion to clarify the unique advantage of FunKAN over other hybrid convolutional KANs. To address this directly, we've designed a new experiment on MRI reconstruction, a task requiring precise spatial processing, specifically, Gibbs-ringing suppression. As shown in Table 4, when integrated into the same hybrid convolutional architecture, replacing an MLP backbone with standard KANs (B-spline, Chebyshev, Hermite) yields only marginal gains (~0.1 dB PSNR). In stark contrast, the proposed FunKAN backbone achieves a substantial +1 dB PSNR improvement. This provides a key methodological insight: the modest gains from other KANs indicate that their fundamental operation - flattening and processing coordinates independently - fails to leverage the inherent structure of spatial image data. FunKAN's success stems from its ability to learn multivariate functional representations that directly operate on structured feature maps, a capability that pure coordinate-based networks lack. This ablation directly solidifies FunKAN's advantage within a hybrid framework.

---

### Official Review · Reviewer_TdKm · 2025-10-29

**Soundness:** 2
**Presentation:** 2
**Contribution:** 2
**Rating:** 2
**Confidence:** 5

**Summary:**

The paper introduces FunKAN, a Kolmogorov-Arnold (KAN) based network for medical image segmentation. Instead of the typical KAN, which works on a 1D sequence, the authors propose to operate directly on 2D feature maps to preserve spatial structure for medical image segmentation. The UNet-style network is evaluated on diverse medical image modalities, including histopatology, ultrasound and colonoscopy.

**Strengths:**

- The paper is well-written and easy to follow.

- The idea of using 2D feature maps instead of flattening for KAN is interesting.

- Experiments are performed on diverse medical imaging modalities.

**Weaknesses:**

- The main idea of the paper is to use KANs for interpretability in the task of medical image segmentation; however, this is just mentioned in the abstract of the paper. There is no clear motivation for why KAN is used here. If it's the interpretability, why didn't the authors include evidence on the interpretability of KANs?

- The datasets used for the experiments have small sizes less than 700 images, whereas larger datasets would better support generalisation claims.

- Experiments in tables 1/2 do not consider very recent medical image segmentation networks, especially transformer-based ones, and are very limited. Also the improvements made are marginal.

**Questions:**

N/A.

**Details Of Ethics Concerns:**

N/A.

---

> ### Author Response · Authors · 2025-11-27
>
> We thank the reviewer for their thoughtful and constructive feedback.
> We are encouraged by the positive comments on the writing and the novelty of using novel functional KANs.
> We have addressed the raised concerns in a revised version of the manuscript, as detailed below.
>
> **1. Lack of interpretability analysis**
>
> The reviewer rightly points out that our motivation for using KANs - interpretability - was not sufficiently demonstrated.
> We have strengthened the interpretability analysis in Experiments section (see Fig. 7).
> A key interpretability advantage of FunKAN lies in its structure (see equation 5): the Hermite decomposition of a learned inner function provides a spectral characterization of its complexity. Spectral characterization is caused by the fact that Hermite functions are the eigenfunctions of the integral Fourier transform. So, a rapid decay of the Hermite coefficients indicates that the model primarily utilizes lower-order polynomials, favoring smoother functions. If high-degree coefficients are significant, it means your model is learning very fine-grained, high-frequency details from the data and such behaviour may be interpreted as a sign of overfitting. Moreover, spectral energy analysis is a direct way to perform feature selection or to validate that the model is focusing on features known to be important from domain knowledge.
>
> **2. Concerns about dataset size and generalization**
>
> We acknowledge the reviewer's concern regarding dataset size.
> While individual datasets are not massive, our core contribution is validated across three diverse modalities (histology, ultrasound, colonoscopy) on common medical imaging benchmarks (BUSI, GlaS, CVC). This demonstrates the robustness, generalizability and SoTA performance of the proposed FunKAN across different medical imaging challenges, which is a significant strength.
> To further ensure FunKAN's generalization across other image processing pipelines, we've applied FunKAN as a backbone for MRI enhancement,  specifically for suppressing Gibbs-ringing after discarding the 75% of high-frequency information on large IXI dataset (>10k training samples) (see Table 4).
>
> **3. Addressing the very recent baselines**
>
> We thank the reviewer for their feedback regarding the baselines. We would like to clarify that our experiments in Tables 1 and 2 specifically include a comparison against U-Mamba, a recently proposed state-of-the-art architecture that is recognized as a leading successor to transformer-based models in medical imaging. By demonstrating clear improvements over this strong, contemporary baseline, we believe our results are both substantial and relevant, precisely addressing the latest architectural advances in the field.

---

### Author Response · Authors · 2025-12-01

Dear Area Chair,

We sincerely thank the reviewers for their thoughtful, constructive and rigorous feedback. The comments have been valuable in strengthening our manuscript. Below, we provide a consolidated summary of our responses and the key revisions made to address the reviewers’ concerns.

**Summary of Major Revisions**

Our revisions focus on two core areas raised across the reviews: **(1) empirical validation of interpretability** and **(2) broader demonstration of generalizability beyond a single task**.

**1. Substantiated Interpretability Claims**

All reviewers highlighted the need to empirically demonstrate the interpretability, promised by our functional KAN (FunKAN) framework. In response, we have added a new interpretability analysis (see Fig. 7) in the Experiments section. This analysis leverages FunKAN design, specifically, the decomposition of learned inner functions using eigenfunctions of integral Fourier transform (i.e. Hermite functions). It gives a spectral characterization of feature complexity:
- Low-order spectral energy indicates smooth, generalizable features.
- High-order spectral energy signals complex, high-frequency patterns that may correlate with overfitting or adversarial vulnerability.

This provides a principled, model-based mechanism for feature ranking and selection, moving our interpretability claims from a theoretical advantage to a demonstrated, practical feature of FunKAN.

**2. Expanded Validation Demonstrating Generalizability**

Reviewers expressed concerns that our initial experiments (focused on binary segmentation) were too narrow to support claims about FunKAN's utility as a multi-purpose spatial medical backbone. To directly address this, we have conducted a new large-scale experiment on a different task and modality:
- Task: MRI reconstruction (Gibbs-ringing suppression), an image-to-image regression task, distinct from pixel-wise classification.
- Dataset: IXI dataset (>10k training samples from T1, T2, and PD MRI modalities).
- Key Result (see Table 4): When integrated as a backbone into the same convolutional architecture, FunKAN achieves a +1 dB PSNR improvement over other KAN backbones (B-spline, ChebyKAN, HermiteKAN), which show only marginal gains (~0.1 dB) upon naive MLP. Additionally, FunKAN also surpasses a classical deringing method.

This proves FunKAN's effectiveness across different medical image processing pipelines (segmentation and enhancement) and modalities (ultrasound, histology, colonoscopy, MRI).
It robustly counters the concern that the proposed method is overfitted to specific segmentation benchmarks.

**Conclusion**

We believe the revised manuscript directly and comprehensively addresses the reviewers' core concerns. We have:
- Empirically validated the interpretability of FunKAN via spectral analysis.
- Demonstrated robust generalizability on a new, large-scale image enhancement task.

We are confident that these significant improvements strengthen the paper's contribution, rigor, and clarity, making it suitable for publication in ICLR 2026.

Thank you for your consideration.

---

### Meta-Review · Area_Chair_VQJu · 2026-01-02

**Summary:**

The reviewers raised several substantive concerns that ultimately outweigh the strengths of the submission. While the paper proposes an extension of Kolmogorov–Arnold Networks (KAN) and demonstrates better performance on selected medical segmentation benchmarks, all the reviewers questioned the clarity of the core contribution. In particular, the paper initially lacked convincing empirical evidence for the claimed interpretability, and although additional analyses were added in rebuttal, it is unconvincing that the theoretical grounding translates into practically meaningful interpretability for clinical tasks. Concerns were also raised about limited generalization, as experiments were largely restricted to small, binary medical segmentation datasets, with insufficient evaluation on multi-class settings, larger benchmarks, or non-segmentation tasks to justify claims of a general-purpose backbone. Several reviewers additionally highlighted marginal or inconsistent performance gains, potential baseline tuning and evaluation fairness issues, and overstated claims of state-of-the-art performance without the level of rigor expected at the conference. These unresolved issues led to the recommendation to reject the paper.

**Reviewer Concerns:**

I think the rebuttal overall is a little bit immature. It did not fully address concerns raised by the reviewers.

**Reviewer Scores:**

After reading the rebuttal, I believe rebuttal will not improve reviewers' scores.

---

### Decision · Program_Chairs · 2026-01-26

Reject